# Blockade of leukemia inhibitory factor as a therapeutic approach to KRAS driven pancreatic cancer

Man-Tzu Wang[1,2], Nicole Fer[3], Jacqueline Galeas[2], Eric A. Collisson[2,4], Sung Eun Kim[2], Jeremy Sharib[5] & Frank McCormick[2,3]

KRAS mutations are present in over 90% of pancreatic ductal adenocarcinomas (PDAC), and drive their poor outcomes and failure to respond to targeted therapies. Here we show that Leukemia Inhibitory Factor (LIF) expression is induced specifically by oncogenic KRAS in PDAC and that LIF depletion by genetic means or by neutralizing antibodies prevents engraftment in pancreatic xenograft models. Moreover, LIF-neutralizing antibodies synergize with gemcitabine to eradicate established pancreatic tumors in a syngeneic, $Kras^{G12D}$-driven, PDAC mouse model. The related cytokine IL-6 cannot substitute for LIF, suggesting that LIF mediates KRAS-driven malignancies through a non-STAT-signaling pathway. Unlike IL-6, LIF inhibits the activity of the Hippo-signaling pathway in PDACs. Depletion of YAP inhibits the function of LIF in human PDAC cells. Our data suggest a crucial role of LIF in KRAS-driven pancreatic cancer and that blockade of LIF by neutralizing antibodies represents an attractive approach to improving therapeutic outcomes.

[1] Tumor Microenvironment Center, UPMC Hillman Cancer Center; Department of Pharmacology and Chemical Biology, University of Pittsburgh, Pittsburgh, PA 15213, USA. [2] Helen Diller Family Comprehensive Cancer Center, University of California, San Francisco, CA 94158, USA. [3] NCI RAS Initiative, Cancer Research Technology Program, Frederick National Laboratory for Cancer Research, Frederick, MD 21702, USA. [4] Department of Medicine/Hematology and Oncology, University of California, San Francisco, CA 94143, USA. [5] Department of Surgery, University of California, San Francisco, CA 94143, USA. Correspondence and requests for materials should be addressed to F.M. (email: Frank.mccormick@ucsf.edu)

D espite being the most frequent oncogene in human cancer, KRAS has so far proven refractory to targeted inhibition. Thus, strategies for targeting KRAS cancers have focused on downstream effectors, such as the MEK/ERK or PI3K/AKT pathways. Combined therapy against these well-established signaling components continues to undergo clinical evaluation, but efficacy has been limited by toxicity[1], incomplete target inhibition, and rebound activation of upstream or parallel signaling pathways. Thus, alternative approaches are needed to target KRAS-mutated cancers effectively.

RAS-induced autocrine cytokine circuits promote cell transformation, tumor cell survival, angiogenesis, and metastasis in multiple types of cancers via activation of NFκB, STAT3, or other pathways[2], but their relevance in KRAS-mutated solid tumors in general, and PDAC in particular, is incompletely understood. Constitutive activation of STAT3 is present in the majority of PDAC cases, and conditional inactivation of STAT3 diminishes the number of premalignant pancreatic lesions, acinar-to-ductal metaplasia, and pancreatic intraepithelial neoplasia in Pdx1-Cre and LSL-KRAS[G12D] mice[3]. The STAT3-regulating IL-6 cytokine family members IL-6 and LIF are upregulated in human PDAC samples in comparison with normal tissue[3]. Moreover, STAT3 has been shown to play a critical role in several types of malignancy, including lung and pancreatic cancer, and targeting the STAT3-signaling pathway has been proposed as a therapeutic strategy[4]. Unexpectedly, STAT3 has recently been demonstrated to have a tumor-suppressive role in KRAS[G12D]-induced lung tumorigenesis[5]. Thus, we sought to determine the roles and specific downstream signaling pathways of these IL6-family cytokine members in oncogenic KRAS-driven pancreatic cancers.

In this regard, we present data showing KRAS regulates expression of LIF in mouse and human pancreatic cancers. Depleting LIF expression by genetic means or blocking its activity by antibody prevents pancreatic tumor initiations and resensitizes cancer cells to Gemcitabine. Furthermore, we show that LIF, but not IL-6, inhibits the activity of the Hippo-signaling pathway in pancreatic cancer cells. Our results suggest that blocking LIF and its specific downstream signaling pathway can provide an alternative approach to improving therapeutic outcomes of pancreatic cancer.

## Results

**Oncogenic KRAS upregulates LIF through the MEK/ERK cascade.** First, we evaluated whether the expression of IL-6-family cytokines is modulated by oncogenic KRAS in pancreatic cancer. Among all IL-6 family members, only LIF mRNA expression decreased in human PDAC cell lines in which oncogenic KRAS had been knocked down by shRNA (Supplementary Fig. 1a). Likewise, expression of LIF protein and activation/phosphorylation of STAT3 were decreased in human and mouse PDAC cells in which KRAS was depleted (Fig. 1a). Adding LIF into the culture medium reactivated STAT3 in cells in which KRAS has been knocked down (Supplementary Fig. 1b). When the expression of KRAS was reactivated in a mouse pancreatic adenocarcinoma cell line (iKRAS*)[6], LIF expression and the sequential activation of STAT3 were significantly increased (Fig. 1b). Furthermore, LIF expression was elevated in human PDAC cell lines, which express higher RasGTP (active RAS) than in those expressing wild-type KRAS/lower RasGTP level (Fig. 1c). The results suggest that the level of LIF expression depends on the levels of active KRAS. LIF, but not IL-6 or IL-11 (reportedly the dominant IL-6-family cytokines in multiple types of cancer[7]), was significantly upregulated in human pancreatic carcinomas when compared with normal pancreas tissue (Supplementary Fig. 1c). In a pan-cancer analysis, LIF expression was greater in

tumor cells expressing mutant KRAS than in those expressing wild-type KRAS, whereas there was no significant difference in IL-6 expression (Fig. 1d). Furthermore, inhibition of MEK, but not AKT, by small molecules downregulated the expression of LIF in human pancreatic cancer cell line Panc1.0, suggesting that the MEK/ERK signaling pathway is essential for regulation of LIF expression by KRAS (Fig. 2a). Adding LIF in the culture media partially rescued its ability to grow in 2D and 3D, which was impaired by MEK inhibition (Fig. 2b).

**LIF mediates malignancies in KRAS-mutant pancreatic cancers.** Next, we examined the functional roles of LIF in oncogenic KRAS-driven pancreatic cancers. We previously demonstrated that oncogenic KRAS is required for human PDAC cells (which are either KRAS-dependent or independent in 2D culture[8]) to grow as spheres in 3D culture[9]. The addition of human LIF to culture media rescued sphere-forming ability that had been suppressed by depletion of KRAS expression in human PDAC cell lines (Fig. 3a), whereas the addition of human IL-6 at the same concentration did not (Fig. 3b). LIF showed more profound effect on increased expression of the stem cell markers, CD44 and abcb1, in human PDAC cell lines in which KRAS had been depleted, when compared to IL-6 (Supplementary Fig. 2a). Moreover, knocking down LIF by shRNA impaired the ability of human pancreatic cancer cell line to grow as spheres in 3D culture. This phenotypic change could be rescued by the addition of LIF, but not IL-6, to the culture medium (Supplementary Fig. 2b). In contrast, knocking down human IL-6 in the same cell line had no effect on its sphere-forming efficiency (Supplementary Fig. 2b). Knocking down LIF by shRNA repressed the tumor initiation and growth rate of human pancreatic cancer cell line in xenograft models (Supplementary Fig. 2b,c). In addition, knockdown of LIF but not IL-6 impaired tumor initiation rate in xenograft models (Supplementary Fig. 2b). Knocking out LIF by CRISPR/Cas9 repressed sphere-forming ability of a mouse pancreatic cancer cell line, which were isolated from a FVB/n Kras[LSL-G12D/+]; Trp53[flox/+]; Ptf1a-Cre mouse[10] (Fig. 3c). Its impaired ability to grow as spheres in 3D can be rescued by the addition of mouse LIF in culture media (Fig. 3c). In addition, the depletion of LIF significantly enhanced overall survival rate in FVB/n mice, which received syngeneic transplantation of pancreatic cancer cells (Fig. 3c).

We next expressed Tet-inducible shRNA against human LIF in multiple human pancreatic cancer cell lines, to study the role(s) LIF plays in KRAS-driven pancreatic cancer maintenance (Fig. 3d, f). Expression of an inducible shRNA against LIF significantly blocked the sphere-forming ability of human PDAC cells in vitro as well as tumor initiation and growth in vivo in a doxycycline-dependent manner (Fig. 3d, e). In the absence of doxycycline, control cells and cells expressing inducible sh-LIF showed comparable tumor initiation and growth rates (Fig. 3f). After introducing doxycycline to mice bearing established xenograft tumors, those expressing inducible shRNA against LIF showed significantly reduced tumor growth when compared with controls (Fig. 3f). Moreover, even though gemcitabine retarded tumor growth, this widely-used chemotherapeutic agent did not lead to meaningful tumor regression in either the control or the shLIF group in the absence of doxycycline (Fig. 3f). In contrast, gemcitabine administered to LIF-depleted tumor-bearing mice induced an improved response, with complete tumor regression in seven of nine animals, whereas there was no significant effect on vehicle control tumors (Fig. 3f). The regressed tumors did not re-grow after the treatments have been withdrawn (Fig. 3f). Moreover, knocking down LIF by shRNA sensitized mouse PDAC cells to multiple chemotherapeutic

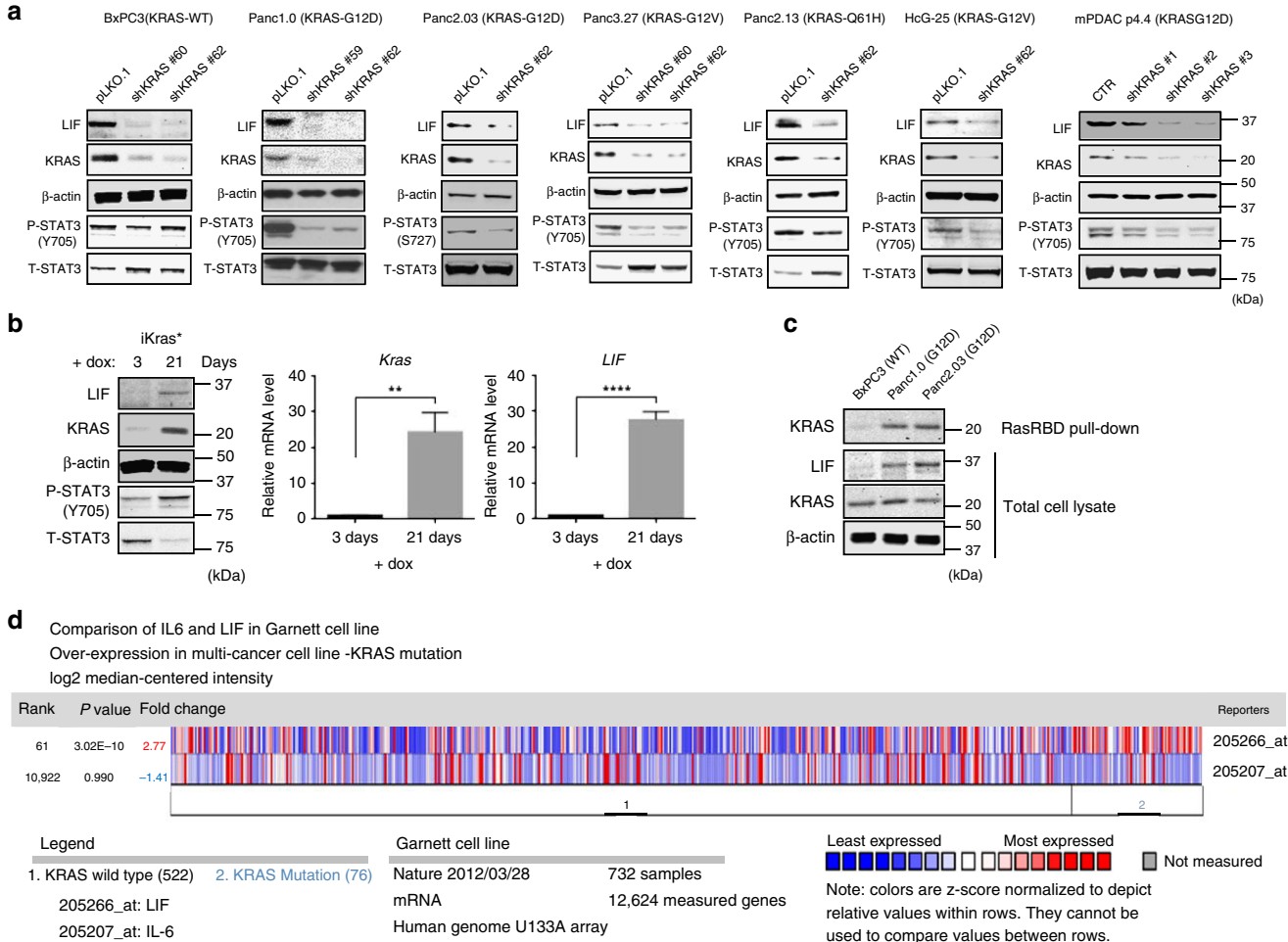

**Fig. 1** Oncogenic KRAS regulates the expression of LIF, but not other IL-6 family cytokines. **a** Representative western blots showing the amounts of LIF, KRAS, and phosphorylated STAT3 at Y-705 in multiple human and mouse pancreatic cancer cell lines where wild-type or mutant KRAS have been stably knocked down by shRNAs. **b** Representative western blots showing increased expression of LIF, KRAS, and phosphorylated STAT3 at Y-705 in mouse pancreatic adenocarcinoma cell line (iKRAS*). Representative qPCR showing increased mRNA expression of LIF and KRAS in the same cell line ($N = 3$; **$P < 0.01$; ****$P < 0.0001$). Error bar represents the standard deviation and $P$-value was generated by $t$-test. Its expression of KRAS was reactivated by adding 1 µg/mL of doxycycline in growth medium. iKRAS* line was harvested and established from pancreatic tumor developed in a p48Cre; TetO-KrasG12D; Rosa26rtTa/+; p53R172H/+mouse. **c** Western bot suggesting increased expression of LIF in KRAS mutant human pancreatic cancer cell lines, Panc1.0 and Panc2.03, which had higher RasGTP level when compared to KRAS wild-type cell line, BxPC3. RasRBD pull-down assay was used to determine RasGTP level. The cells were serum-starved for 24 h before harvested for RasRBD pull-down. **d** Oncomine analysis suggesting upregulation of human LIF, but not IL-6, in the human cancer cell lines where KRAS is mutant when compared with cells expressing wild-type KRAS.The analysis in the pan-cancer cell lines (KRAS WT vs mutant) (*Nature* **volume 483**, pages 570–575 (29 March 2012)) was performed by using Oncomine Platform Software (Invitrogen; https://www.oncomine.org/resource/login.html)

agents, including cisplatin, gemcitabine, and 5-FU (Supplementary Fig. 2d). Expression of LIF protein was increased in human PDAC cells, which were resistant to 5-FU in vitro (Supplementary Fig. 2e). These results indicate that LIF plays a crucial role in tumor maintenance and the sensitivity of pancreatic cancer to chemotherapy.

**Blocking LIF activity suppresses pancreatic malignancies.** Next, we tested whether a monoclonal antibody (D25.1.4) specifically targeting LIF has therapeutic potential in KRAS-driven pancreatic models (Fig. 4a, b). Blocking LIF activity by antibody inhibited the sphere-forming ability of mouse pancreatic cancer cells (Supplementary Fig. 3). Treatment with LIF-neutralizing antibody at 10 mg/kg/mouse before injection of human pancreatic cancer cells, followed by treatment twice a week, successfully prevented tumor establishment in a pancreatic xenograft model

(Fig. 4c). We then evaluated whether LIF-neutralizing antibody can improve therapeutic outcomes when combined with conventional chemotherapy. In syngeneic mouse models bearing oncogenic Kras$^{G12D}$-driven pancreatic adenocarcinoma, gemcitabine or antibody treatment alone showed no significant effects on tumor growth (Fig. 4d). In contrast, LIF-neutralizing antibody along with gemcitabine significantly repressed tumor growth and increased overall survival (Fig. 4e). To estimate the effect of inhibiting LIF in a model, which reflects tumor heterogeneity in patients, we next tested LIF-neutralizing antibody in patient-derived xenografts (PDXs), which expressed detectable LIF protein (Fig. 4f, g, Supplementary Fig. 4). LIF antibody along with gemcitabine significantly repressed tumor growth in a PDX model (Fig. 4f, g). Those tumors showed no sign of re-growth after treatment, whereas gemcitabine-treated tumors recurred after the treatment had been withdrawn (Fig. 4f).

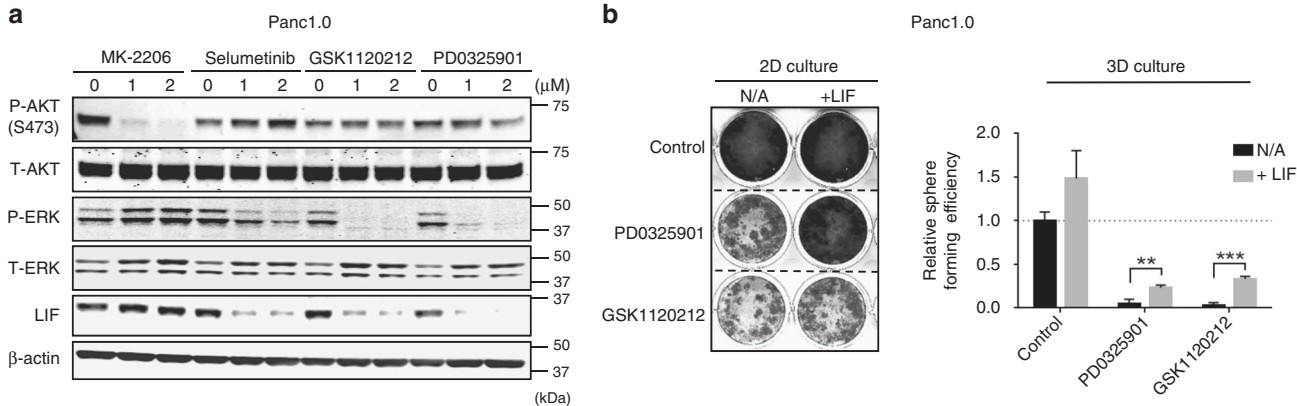

**Fig. 2** Inhibition of MAPK-signaling pathway represses expression of LIF. **a** Representative western blots showing the amounts of LIF, phosphorylated ERK and phosphorylated AKT in Panc1.0 cells. Inhibition of the MAPK-signaling pathway by small molecules selumetinib, GSK-1120212, and PD0325901 suppressed the expression of LIF, whereas inhibiting the activity of AKT by MK-2206 did not. The inhibitory efficiency of the small molecules was confirmed by the phosphorylation levels of AKT at S-473 and the phosphorylation of ERK. The cells were treated with compounds for 48 h before harvested for western blot analysis. **b** Growth of Panc1.0 in 2D and 3D culture suppressed by MEK inhibitors, GSK-1120212, or PD0325901 at 1 µM. The suppression of growth can be rescued by adding LIF in culture medium. Crystal violet staining was used to visualize cell growth in 2D culture. Sphere formation assay was used to determine the growth in 3D ($N = 6$; **$P < 0.01$; ***$P < 0.001$). Error bar represents the standard deviation and $P$-value was generated by $t$-test

**LIF suppresses the HIPPO pathway**. Our results suggest that, even though IL-6 and LIF share STAT3 as a downstream effector, they have distinct functional roles in human pancreatic cancer cells harboring mutant KRAS. Furthermore, IL-6 showed a stronger ability to activate/phosphorylate STAT3 at Tyr705 than did LIF or another IL-6 family member, IL-11, in human pancreatic cancer cells (Supplementary Fig. 5a). These observations raise the possibility that LIF could mediate KRAS-driven pancreatic malignancies through a non-STAT3-signaling pathway. Binding of LIF to the LIFR-gp130 receptor complex has been previously shown to activate YES and YAP/TAZ-TEAD -dependent transcription, which is required to maintain self-renewal in embryonic stem cells[11]. The YES-associated protein (YAP) and its transcriptional co-activator, TEAD or TAZ, are negatively regulated by the Hippo-signaling pathway[12]. Aberrant activation of YAP and TAZ due to deregulation of the Hippo pathway or overexpression of YAP/TAZ and TEADs can promote cancer development[10,11].

To understand whether LIF and IL-6 have different effects on the activation of YAP/TAZ-TEAD in pancreatic cancers, we treated serum-starved human pancreatic cancer cells with IL-6 or LIF at 100 ng/mL. In this model, treatment with LIF, but not IL-6, significantly elevated YAP/TAZ-TEAD transcriptional activity and sequentially increased expression of YAP1-targeted genes, *CNTF* and *ANKRD* at mRNA, in human PDAC cells (Fig. 5a and Supplementary Fig. 5b). In addition, knocking down LIF by inducible shRNAs reduced YAP-TEAD/TAZ transcriptional activity or nucleus- localized YAP in human PDAC cell lines (Fig. 5b, c). Doxycycline-induced expression of shRNA against LIF increased phosphorylation of YAP at Ser127, which caused its cytoplasmic retention for subsequent degradation[13,14], and activated multiple Hippo-signaling pathway components, including LATS1 and MOB1, in human pancreatic cancer cells[15] (Fig. 5d). Similarly, knocking out LIF by CRISPR/Cas9 enhanced phosphorylation of YAP at Ser127 in mouse pancreatic cancer cells (Supplementary Fig. 5b). Treatment with LIF antibody had the same effects on activation of Hippo-signaling pathway in human pancreatic cancer cells without altering their KRAS expression and ERK activity (Fig. 5e). Furthermore, tumors from PDX or syngeneic tumors receiving LIF-neutralizing antibody alone or along with gemcitabine had increased phosphorylation of YAP at Ser127 when compared with samples from control or tumors treated with gemcitabine only (Fig. 5f and Supplementary Fig. 5e). In the same tumors, treatment with LIF antibody reduced the level of phosphorylation of STAT3 at Y705, whereas gemcitabine treatment alone increased the phosphorylation/activation of STAT3 (Fig. 5f).

To validate the role of YAP/TAZ-TEAD transcription in LIF-mediated pancreatic malignancies, we used siRNA or shRNA to disrupt the interaction between YAP and TAZ[16]. Multiple pancreatic cancer cell lines where LIF had been knocked down were first seeded in 3D culture with or without LIF, and viable spheroid cells were quantified by CellTiter Glo assay on the fourteenth day. As expected, the presence of LIF enhanced the number of viable spheroid cells derived from human pancreatic cancer cells when compared with controls (Fig. 5g and Supplementary Fig. 5d). Knocking down YAP significantly compromised their ability to grow as spheres in 3D culture, which failed to be rescued by addition of LIF in culture medium (Fig. 5g and Supplementary Fig. 5d). Furthermore, histochemical staining revealed that expressions of LIF and phospho-YAP at Ser127 are negatively correlative in human pancreatic tissues (Biomax human pancreatic cancer tissue array PA484 and PAN241) (Fig. 5h). These results suggest that LIF mediates KRAS-driven pancreatic cancer through suppression of the Hippo pathway, which subsequently increases YAP/TAZ-TEAD transcriptional activity. In addition, the phosphorylation of YAP at Ser127 might be deployed clinically as a predictive biomarker to indicate the activation of the LIF-signaling pathway.

## Discussion

With over 90% pancreatic adenocarcinomas harbor oncogenic mutations in KRAS, identification of downstream effectors essential for KRAS-mediated tumorigenicity and drug resistance can lead to new targets of intervention for this intractable disease. Our data indicate that LIF is stimulated by KRAS and further it plays an important role in facilitating KRAS to drive pancreatic cancer. Neutralization of LIF is found to diminish or even block the oncogenic capability of KRAS and reduce PANC resistance to chemotherapy.

LIF is a member of IL-6 superfamily of cytokines, which include oncostatin M (OSM), IL-6, IL-11, ciliary neurotrophic

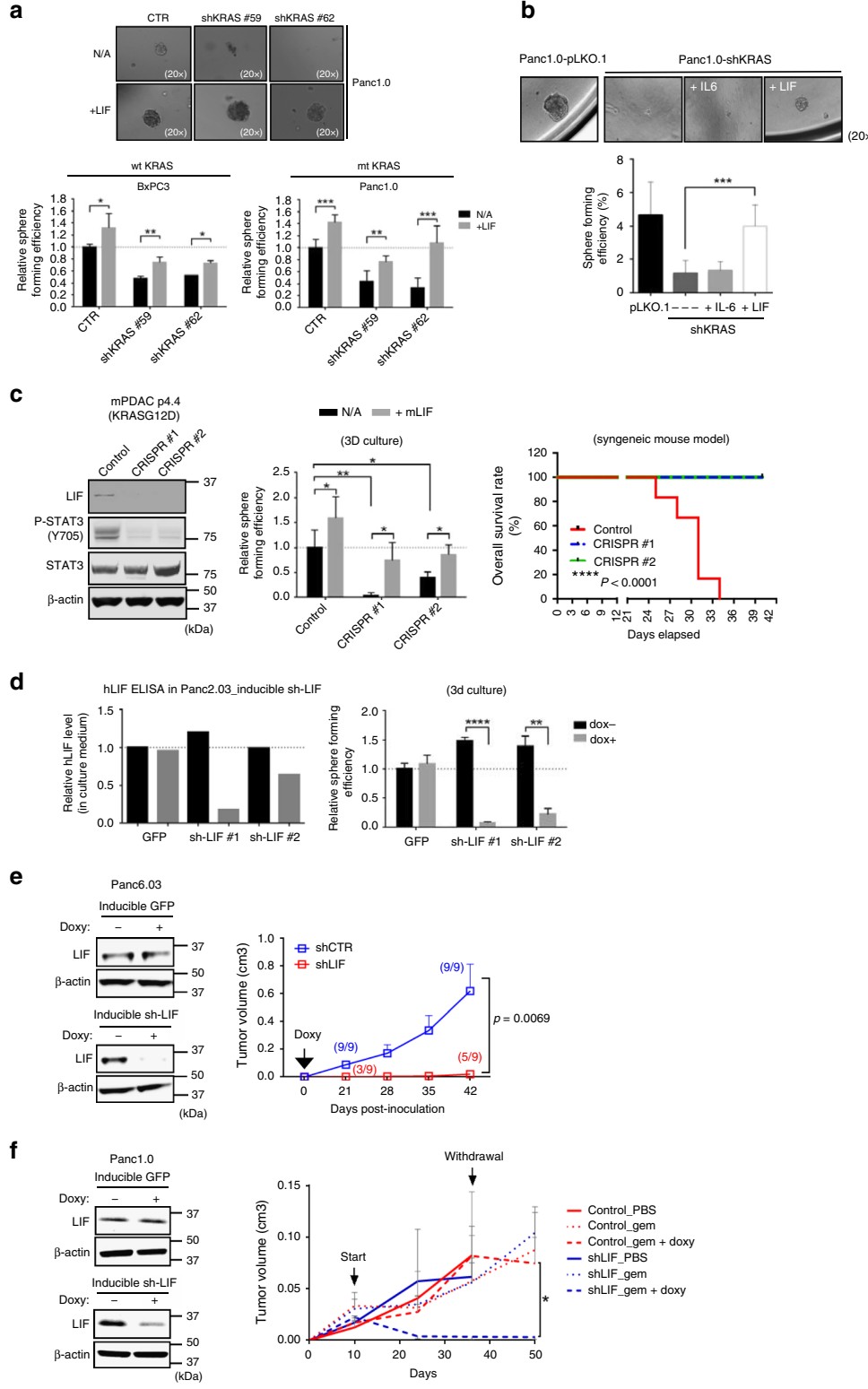

factor (CNTF), and cardiotrophin-1 (CT-1). LIF initiates cell signaling through formation of heterodimer with specific LIFR and the common co-receptor for IL-6 family (gp130). Signal transduction of IL-6 family members is highly similar and dominated by STAT3 activation. Aberrant/unrestrained STAT3 activity is detected in a wide variety of tumors, including pancreatic and lung adenocarcinomas[4]. Thus, STAT3 has been widely considered as an oncogene and is the object of intense translational studies. Yet, none of the anti-STAT3 agents, ranging from small-molecule inhibitors to blocking its upstream cytokines, has demonstrated efficacy in clinical trials. Recent studies suggest that STAT3 exhibits either pro-oncogenic or tumor-suppressive activity depending on the tumor aetiology/mutational landscape. For example, STAT3 is frequently activated and plays oncogenic roles in non-small cell lung adenocarcinomas with the context of *EGFR* driver mutations, whereas low STAT3 level correlate with increased malignant progression and poor prognosis in lung cancer patients with *KRAS* mutations. In addition,

**Fig. 3** Expression of LIF is required for in vitro and in vivo malignancies of human pancreatic cancer cells. **a** The ability to form spheres in 3D culture, reduced by the depletion of KRAS, could be rescued by addition of LIF at 100 ng/mL. Sphere size is shown in the left panel, and sphere-forming efficiency (numbers of formed spheres/numbers of seeded cells) in the right panel ($N = 6$, *$P < 0.05$, **$P < 0.01$, ***$P < 0.001$; ****$P < 0.0001$). **a** The ability of Panc1.0 to form spheres in 3D culture, reduced by knocking down KRAS, could be rescued by the addition of LIF, but not IL-6 at the same concentration (100 µg/mL) ($N = 6$, ***$P < 0.001$). **b** Representative western blots (left panel) showing the depletion of LIF protein expression by CRISPR/Cas9 and reduced phosphorylated STAT3 at Y705 in mouse pancreatic tumor cells. The ability to form spheres in 3D culture (middle panel), reduced by depletion of LIF, can be rescued by addition of mouse LIF at 100 ng/mL ($N = 6$, *$P < 0.05$, **$P < 0.01$). (Right panel) Overall survival rate of syngeneic mice which were orthotopically injected with mouse pancreatic cancer cells ($n = 5$). **c** Representative ELISA (left panel) detecting human LIF in culture medium of Panc2.03, which expressed inducible shRNA against human LIF. The cells were treated with doxycycline (1 µg/mL) for 14 days before ELISA assay. Reduced sphere formation efficiency (right panel) of cells where LIF had been knocked down by inducible sh-LIF in the presence of doxycycline ($N = 6$, **$P < 0.01$, ****$P < 0.0001$). Error bars from **a** to **d** represent the standard deviation and $P$-value was generated by $t$-test. **d** Representative western blots (left panel) showing LIF protein expression in Panc6.03 cell line which expressed inducible shRNA against human LIF. Induced expression of shRNA against LIF by doxycycline reduced the growth of the established Panc6.03 xenograft tumors when compared with the control (right panel). **e** Representative western blots (left panel) showing LIF protein expression in Panc1.0, which expressed inducible shRNA against human LIF. Tumor growth curve (right panel) of Panc1.0 xenografts where inducible LIF shRNA was expressed in response to gemcitabine at 50 mg/kg with or without doxycycline

conditional deletion of STAT3 in mouse lung epithelial cells increases carcinogen or oncogenic KRAS-induced tumorigenesis[5]. In the present studies, IL-6 is a stronger activator of STAT3 than LIF, but unlike LIF, IL-6 could not rescue or restore the sphere-forming abilities of PANC as result of KRAS knockdown. Neutralization of LIF in PANC tumors led to decreased phosphorylation/activation of STAT3 at Y705, but gemcitabine treatment increased it. The data suggest the complexity of STAT3 in tumorigenicity and treatment responses of cancers with oncogenic KRAS.

Among members of IL-6 family, LIF is unique in its ability to regulate self-renewal of stem cells through activation of YES and YAP/TAZ-TEAD-dependent transcription[11]. We found LIF, but not IL-6, suppressed Hippo pathway in human PDAC cells, as evidenced by increased YAP/TAZ-TEAD transactional activities and increased expression of YAP1-targeted genes *CNTF* and *ANKRAD*. Neutralization of LIF or LIF knockdown led to increased phosphorylation of YAP at Ser127 in both human and mouse pancreatic cancer cell lines. YAP is critical for LIF activities since if YAP is knocked down, LIF could not promote spheroid formation. In human pancreatic tumor specimens, LIF is negatively correlated with phospho-YAP at Ser127, suggesting the clinical relevance of LIF-YAP signaling circuitry identified.

Although LIF is known to regulate the embryonic stem cell self-renewal and is an indispensable factor to maintain mouse embryonic stem cell pluripotency, LIF also plays an important role in embryonic implantation, and such functions cannot be replaced by other members in IL-6 family. Levels of IL-6 are very low under normal conditions but can elevate thousand-fold in inflammatory states. Elevated IL-6 levels are used to characterize autoimmune diseases, such as rheumatoid arthritis and inflammatory bowel disease. In addition, IL-6 is necessary and sufficient to reverse human T-cell suppression by Treg while anti-IL-6 restored Treg-mediated suppression. By contrast, LIF regulates development and proliferation of Treg by suppressing IL-6-induced Th17 lineages development. Further studies are needed to determine whether LIF and IL-6 counter-regulate development of T cell lineages in the microenvironment of cancers.

In conclusion, our results indicate that disrupting LIF signaling, which directly fuels oncogenic KRAS-driven pancreatic adenocarcinoma, has an impact on tumorigenicity of KRAS and overcomes the characteristic resistance to chemotherapy. Disruption of the autocrine LIF circuit using neutralizing antibody may be a promising new therapeutic approach for KRAS tumors. Identifying non-STAT3 downstream effectors/signaling pathways specifically mediated by LIF can be beneficial for evaluating therapeutic efficiency of LIF blockade in *KRAS* mutant cancers.

## Methods

**Cell lines.** Human pancreatic cancer cells were purchased from ATCC. BxPC3 and SW1990 were grown in RPMI-1640 medium supplemented with 10% FBS. Panc1.0, Panc2.13, HcG25 and mouse tumor cell lines were grown in Dulbecco's modified Eagle's medium (DMEM) supplemented with 10% FBS. Panc2.03 and Panc3.27 were maintained in ATCC-modified RPMI-1640 medium supplemented with 15% FBS and human recombinant insulin (Gibco 12585-014). All the cells were maintained at 37 °C, 5% $CO_2$.

**Antibodies.** LIF antibody (LSBio, LS-B7078-0.05) for IHC (1:500 dilution). LIF antibody (abcam, ab34427) for western blotting (1:500 dilution). KRAS antibody (Sigma–Aldrich, WH0003845M1) for western blotting (1:500 dilution). β-Actin antibody (Sigma–Aldrich, A5441) for western blotting (1: 10,000 dilution). STAT3 (phospho Y705) antibody (abcam, ab76315) for western blotting (1:1,000 dilution). STAT3 (phospho S727) antibody (Cell Signaling, #9134) for western blotting (1:1,000 dilution). STAT3 antibody (Cell Signaling, #9139) for western blotting (1:1,000 dilution). AKT (phospho S473) antibody (Cell Signaling, #4060) for western blotting (1:1,000 dilution). AKT antibody (Cell Signaling, #4691) for western blotting (1:1,000 dilution). P44/42MAPK (Erk1/2) antibody (Cell Signaling, #4695) for western blotting (1:2,000 dilution). Phospho-p44/42MAPK (Erk1/2) (Thr202/Tyr204) antibody (Cell Signaling, #9106) for western blotting (1:2,000 dilution). Phospho-LATS1 antibody (Cell Signaling, #9157) for western blotting (1:1,000 dilution). LATS1 antibody (Cell Signaling, #3477) for western blotting (1:1,000 dilution). Phospho-MOB1 antibody (Cell Signaling, #8699) for western blotting (1:1,000 dilution). MOB1 antibody (Cell Signaling, #13730) for western blotting (1:1,000 dilution). YAP (Phospho S127) antibody (Cell Signaling, #4911) for western blotting (1:1,000 dilution). YAP antibody (Cell Signaling, #14074) for western blotting (1:1,000 dilution). YAP (Phospho S127) antibody (abcam, ab76252) for IHC (1:200 dilution). GAPDH antibody (Trevigen, 2275-PC-100) for western blotting (1:1,000 dilution).

**Animal studies.** All experiments were approved by the IACUC of the University of California, San Francisco. Human pancreatic cancer cells were subcutaneously injected into female nude mice (Nu/Nu) at 0.1 or $1 \times 10^6$ cells per flank. Tumors from PDXs were cut into even size piece ($4 \times 2$ mm tissue fragments) and implanted subcutaneously in 6-weeks-old SCID mice (Charles River Laboratories International, catalog number: 236). Palpable tumors were measured twice a week. The animals were divided into at least five mice per group. Pancreatic adenocarcinoma cells were derived from $Kras^{LSL-G12D}$ mice, genotyped as described[15]. One hundred cells were orthotopically implanted into 6- to 8-week-old FVB/n mice in 20 µL 50% Matrigel with a 28.5-gauge needle. Mice were monitored for one month and euthanized when distressed. Histologically confirmed skin carcinomas were analyzed for DNA, RNA and protein by conventional methods. All studies were conducted in accordance with the UCSF Institutional Animal Care and Use Committee, and all relevant ethical regulations were followed.

**RNA interference.** The shRNAs vectors targeting KRAS, LIF, IL-6, and YAP were purchased from Open Biosystems. The shRNA constructs were packaged as lentiviruses using third-generation lenti-virus packaging systems with standard protocols. The packaging plasmids were from Addgene. Puromycin (final concentration = 2 µg/mL) was used to select infected cells. Clone IDs: KRAS TRCN0000033259-62. Human LIF TRCN0000058584-86. Human IL-6 TRCN0000059205-07.

**CRISPR/Cas9.** Sanger CRISPR QuickPick™ Knockout Clones MM5000026569 (targeting mouse LIF gene sequence TTCTGGTCCCGGGTGATATTGG) and MM5000026570 (targeting mouse LIF gene sequence

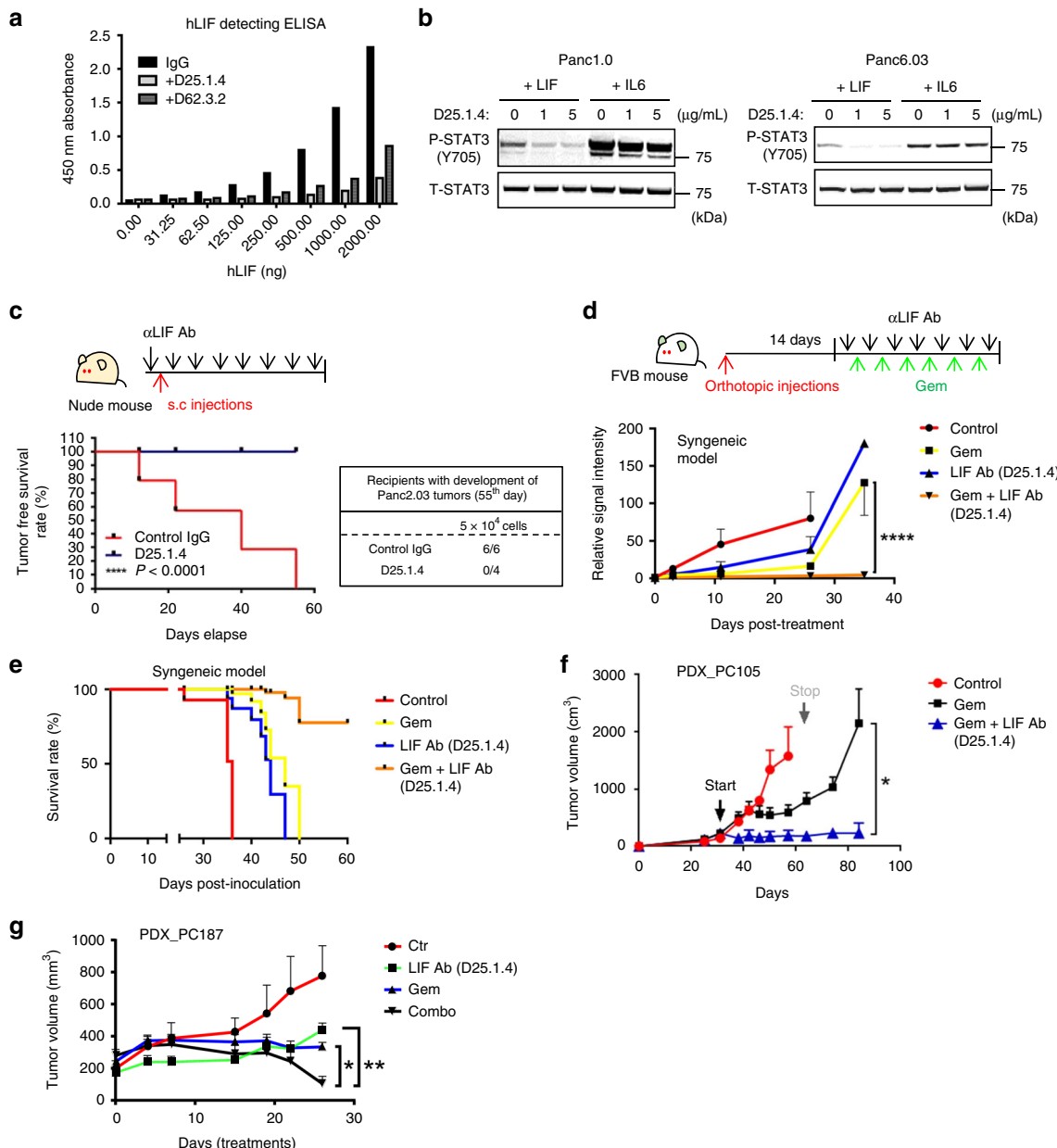

**Fig. 4** LIF-neutralizing antibody acts as a therapeutic approach for treating KRAS-driven pancreatic cancers. **a** Representative ELISA assay showing that LIF antibodies (D25.1.4 and D62.3.2) prevent recombinant human LIF from binding to microplate pre-coated with human LIF monoclonal antibody (Invitrogen BMS242). **b** Representative western blots suggesting that use of a LIF-neutralizing antibody blocks the STAT3 activation/phosphorylation induced by recombinant human LIF (50 ng/mL) in human pancreatic cancer cell lines, whereas IL-6 induces STAT3 activation. **c** Tumor free survival rate and numbers of palpable tumors in Panc2.03 xenografts model receiving control or LIF antibody at 10 mg/kg. (**d**) Tumor growth in syngeneic mouse model receiving gemcitabine (50 mg/kg), LIF antibody (20 mg/kg), or gemcitabine along with LIF antibody. The cancer cells are expressed with firefly luciferase, which can be used to detect orthotopic tumor growth. Twice a week of treatments started on the fourteenth day post-tumor cell injections ($N = 5$, ****$P < 0.0001$). **e** Overall survival rate of syngeneic mouse model receiving gemcitabine, LIF antibody, or gemcitabine along with LIF antibody. **f** Tumor growth curve of PDX (patient-derived xenograft)_PC105 receiving gemcitabine (100 mg/kg) or gemcitabine along with LIF antibody (10 mg/kg). The treatment of gemcitabine was conducted once a week and LIF antibody was authorized twice a week ($N = 6$, *$P < 0.05$). **g** Tumor growth curve of PDX_PC187 receiving control IgG, gemcitabine (100 mg/kg), LIF antibody (10 mg/kg), or gemcitabine along with LIF antibody (10 mg/kg). The treatment of gemcitabine was conducted once a week and LIF antibody was authorized twice a week ($N = 7$, *$P < 0.05$; **$P < 0.01$). **d**, **f**, **g** Error bars: mean ± SEM and $P$-value was generated by Two-way ANOVA

TTGGTGGAGCTGTATCGGATGG) were purchased from Sigma–Aldrich. The backbone vector is U6-gRNA; hPGK-puro-2A-tBFP. The constructs were expressed in mouse pancreatic cancer cells. The knocking out efficiency was determined by western blot analysis.

**Western blot analysis**. Experimental cells were washed twice in ice-cold PBS and lysed in 1% Triton lysis buffer (25 mmol/L Tris pH 7.5, 150 mmol/L NaCl, 1%

Triton X-100, 1 mmol/L EDTA, 1 mmol/L EGTA, 20 mmol/L NaF, 1 mmol/L $Na_2VO_4$, and 1 mmol/L DTT) supplemented with a protease inhibitor cocktail (Roche) and cleared by centrifugation. Protein concentrations were determined by the Bio-Rad Protein Assay (Bio-Rad). Equal amounts of protein extracts were resolved with SDS-PAGE (NuPAGE; Invitrogen), transferred to a nitrocellulose membrane and immunoblotted with primary antibodies indicated, followed by secondary antibodies, labeled with either IRDye800 (Rockland) or Alexa Fluor 680 (Molecular Probes), and visualized with a LI-COR Odyssey scanner.

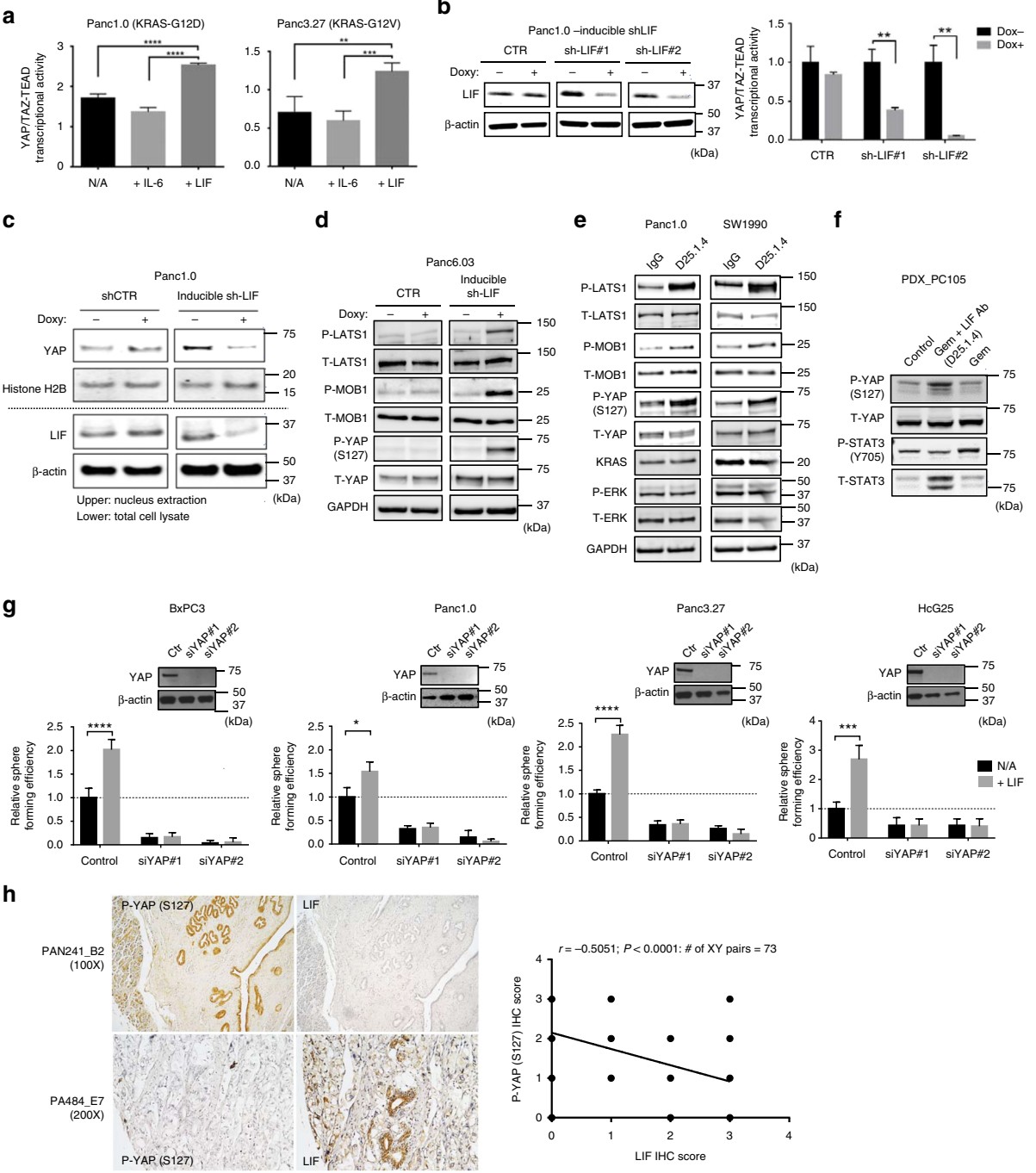

**RasGTP pull-down assay**. Cells were washed twice in ice-cold PBS and lysed in 1% TX100-TNM lysis buffer (20 mmol/L Tris pH 7.5, 5 mmol/L MgCl2, 150 mmol/L NaCl, 1% Triton X-100) supplemented with 1 mmol/L DTT, and protease and phosphatase inhibitors (Sigma–Aldrich). Equal amounts of protein from each sample were added to 10 µL of packed GST-Raf-RBD or Ral-GDS-RBD beads in 300–500 µL of 1% TX100-TNM lysis buffer and rotated at 4 °C for 1–2 h. Beads were washed three times with 1 mL of cold lysis buffer and boiled in lithium dodecyl sulfate (LDS) sample buffer (Invitrogen).

**Sphere formation**. Cells were harvested, counted and seeded onto Ultra Low Attachment Culture 96-well plates (Corning Life Science, Catalog number 3261) at 100 or 1000 per well. The seeded cells and formed spheres were maintained in low-serum-containing medium with 0.1% FBS. The initiated spheres were observed twice a week, with the number of formed spheres counted one month after seeding.

**LIF ELISA**. The level of human LIF in culture medium was determined by human LIF ELISA kit (Invitrogen Cat. No. BMS242). An anti-human LIF antibody was

pre-coated onto microwells. Human LIF present in the sample or standard binds to antibodies adsorbed to the microwells. Following incubation unbound biological components are removed during a wash step. A biotin-conjugated anti-human LIF antibody is added and binds to human LIF captured by the first antibody. Following incubation unbound biotin- conjugated anti-human LIF antibody is removed during a wash step. StreptavidinHRP is added and binds to the biotin-conjugated anti-human LIF antibody. A colored product is formed in proportion to the amount of human LIF present in the sample or standard. The reaction is terminated by addition of acid and absorbance is measured at 450 nm.

**Crystal violet staining**. The cells were seeded into six-well plates at $0.1 \times 10^6$ cells per well with complete growth medium in the absence or presence of doxycycline. On the ninth day, the colonies were stained and visualized by 0.05% crystal violet staining (in 0.1% methanol).

**Quantitative PCR**. Total RNAs were isolated and purified with the QIAGEN RNAeasy kit; 1 µg RNA per specimen was reverse-transcribed into cDNA with the

**Fig. 5** LIF suppresses the Hippo-signaling pathway in human pancreatic cancer cells. **a** Representative HOPFlash luciferase assay showing YAP/TAZ-TEAD transcriptional activity in human pancreatic cancer cells receiving IL-6 or LIF at 100 ng/mL ($N = 3$, **$P < 0.01$, ***$P < 0.001$, ****$P < 0.0001$). **b** (Left panel) Representative western blots showing inducible knockdown of LIF in Panc1.0 by using two different hairpin sequences. (Right panel) Representative HOPFlash luciferase assay showing YAP/TAZ-TEAD transcriptional activity in Panc1.0 cells, which express inducible shRNA targeting human LIF ($N = 3$, **$P < 0.01$). Error bars from (**a, b**) represent the standard deviation and $P$-value was generated by $t$-test. **c** Western blots showing subcellular (nuclear) localization of total YAP in Panc1.0 cell line which express inducible shRNA against human LIF. **d** Representative western blots showing protein expression of phosphorylated LAST1, phosphorylated MOB1, and phosphorylated YAP at S127 in Panc6.03 which express inducible shRNA targeting human LIF. **e** Representative western blots showing protein expression of phosphorylated LAST1, phosphorylated MOB1, phosphorylated YAP at S127, KRAS, and phosphorylated ERK in human pancreatic cancer cells treated with LIF-neutralizing antibody at 2 µg/mL. **f** Western blots suggesting expression of phosphorylated YAP at S127 and phospho-STAT3 at Y705 in PDX tumors receiving gemcitabine or gemcitabine along with LIF antibody. The tumor samples were harvested from in vivo assay shown in Fig. 3g. **g** Sphere-forming efficiency in multiple human pancreatic cancer cell lines where YAP is knocked down by using siRNA in the presence or absence of human LIF in culture medium ($N = 6$, *$P < 0.05$, ***$P < 0.001$, ****$P < 0.0001$). Error bar represents the standard deviation. $P$-value was generated by $t$-test. Western blots in the upper panel showing YAP expression in different cell lines. **h** (Left panel) IHC suggesting the expression of LIF or phospho-YAP at S127 in human pancreatic tissues, including normal tissues and malignant tumors (Biomax PA484, PAB241). (Right panel) The correlation curve suggested that the expressions of LIF and phospho-YAP at S127 are negatively correlated in human pancreatic tissues. The imagines were taken, and the intensity of was quantified by the KEYENCE BZ-X800 microscope. The Pearson's correlation coefficient was used to analyze the relationship between the staining index of LIF and phospho-YAP (S127)

---

SuperScript™ First-Strand Synthesis System for RT-PCR (Invitrogen). Possible contamination of genomic DNA was excluded by treatments of DNase I. Quantitative real-time PCR array analysis was performed with SYBR Green (Applied Biosystem). Fold differences and statistical analyses were calculated with the GraphPad Prism 4.00 for Windows (GraphPad Software). The oligos used for qPCR are listed in Supplementary Data 1.

**Luciferase assay**. Cells were transfected with HOP-Flash (Addgene#83467) luciferase reporter construct along with pRL-CMV-*Renilla luciferase* control reporter vector by Fugene6 (Roche). Luciferase activity was measured with Dual Luciferase Reporter Assay System (Promega; #E1910) according to manufacturer's instructions. The reporter's firefly luciferase activity was normalized to the levels of *Renilla* luciferase used as an internal control reporter. The relative luciferase activity displayed on the *Y*-axis indicates the ratio between Firefly/Renilla luciferase activities.

**Immunohistochemistry**. The deparaffinized tissues were unmasked with Cell MarqueTM Trilogy reagent (ALS) in an electric rice cooker for 30 min. The slides were quenched by placement in $H_2O_2$/Methanol for 10 min at room temperature. Human and mouse pancreatic tissues were stained with the Histostain® SP kit (Invitrogen), according to the manufacturer's instructions. The dilution of primary antibodies was done in accordance with the product application note.

**Nucleus extraction**. The cells stably expressing inducible shRNA against LIF were cultured in doxycycline-containing complete medium for at least 10 days prior nucleus extraction. The nuclei were isolated by using Nuclear Extract Kit (Active Motif, cat. no. 40010), according to the manufacturer's instructions. Western blot analysis was followed to evaluate the subcellular protein localization. Nuclear protein Histone H2B was probed and used as a loading control.

**Statistical analysis**. Data from quantitative PCR (qPCR) experiments, sphere formation assay, and luciferase assay were analyzed by two-tailed $t$-test in Excel. All error bars in these assays indicate s.d. of three–six technical replicates, and all the in vitro studies have been performed in three independent experiments. The differences were considered to be statistically significant for $P$-values <0.05 (*), <0.01 (**), <0.001 (***), and <0.0001 (****). Nolinear fit correlation was analyzed in GraphPad Prism. Tumor surface was measured every week with the formula $A = \pi (dD/2)$, where $d$ is the minor tumor axis and $D$ is the major tumor axis. Error bars in the animal studies denoted s.e.m., analyzed in GraphPad Prism. GraphPad Prism was also used for plotting all figures.

**Reporting summary**. Further information on research design is available in the Nature Research Reporting Summary linked to this article.

## Data availability

The data that support the findings of this study are available from the authors on reasonable request, see author contributions for specific data sets. The ONCOMINE reanalysis that support the findings of this study are publicly available online at https://www.oncomine.org/resource/login.html. The accession number linked to Fig. 1 is E-MTAB-783 (ArrayExpress). The accession code linked to Supplementary Fig. 1 is GSE16515. The uncropped images of key western blots are available as the Source Data.

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

## Acknowledgements

This work was supported in part by a grant from the Lustgarten Foundation and by NIH/NCI R01CA178015 and R01CA222862.

## Author contributions

M.-T.W. and F.M. conceived the project, designed the experiments, interpreted the results, and wrote the manuscript. E.A.C. revised the manuscript. M.-T.W. and J.G. performed most of the experiments, with the exception of those in Fig. 4, most of which were performed by N.F. S.-E.K. generated the cell lines where YAP had been knocked down. J.S. and E.A.C. contributed and maintained pancreatic cancer patient-derived xenografts.

## Additional information

**Competing interests:** The authors declare no competing interests.

