## [Peer Review File · Nature Communications]

Editorial Note: This manuscript has been previously reviewed at another journal that is not operating a transparent peer review scheme. This document only contains reviewer comments and rebuttal letters for versions considered at Nature Communications. Mentions of prior referee reports have been redacted.

Reviewers' comments:

Reviewer #1 (Remarks to the Author):

The authors have significantly improved their manuscript in the revised version. Important changes include the addition of more cell lines, two PDX models for the in vivo experiments, and additional knockdown experiments interfering with YAP/TAZ TEAD that suggest a role for this pathway in LIF-mediated signaling, at least in a sphere forming assay. A number of additional improvements have been made to the manuscript to address the reviewers' comments. This includes a much better description of the experimental procedures, improved clarity of the figures and additional minor changes. In general, this is an interesting manuscript in an important field and the experiments appear to support the conclusions. I would recommend publication of the manuscript once the remaining points are addressed. The main point I still consider insufficiently addressed is the description of the statistical parameters used. This omission makes interpretation of some of the experiments difficult or impossible.

Statistical parameters:

Despite having ticked all 'confirmed' boxes on the Reporting Summary form, the authors insufficiently not at all or describe a number of important statistical parameters important for interpretation of the results. Maybe the authors could enlist the help of someone experienced in biostatistics and re-evaluate all figures that include (or should include) statistical testing?

Examples are:

A) The authors should state whether the 'N' indicated refers to biological or technical replicates in cases where this is not clear from the description.

B) The tests used should be indicated and whether they are one- or two-sided.

C) Still, the error bars are not clearly defined.

D) A full description of the statistics including central tendency (e.g. means) or other basic estimates (e.g. regression coefficient) AND variation (e.g. standard deviation) or associated estimates of uncertainty (e.g. confidence intervals) is missing.
and/or

E) For null hypothesis testing, the test statistic (e.g. F, t, r) with confidence intervals, effect sizes, degrees of freedom and P value noted

Reviewer #2 (Remarks to the Author):

This manuscript, which has been previously reviewed for [redacted] has been significantly improved with more in vivo models and genetic YAP inactivation.

In principal I think that this work is suitable for acceptance, however, some issues remain.

- Fig. 1d+e: the legends got mixed up, please revise. The "Oncomine analysis" remains ill-described and should include more details to follow it.
- Fig. 3g: the authors explicitly state (pg. 6, line 126) that anti-LIF was tested in PDXs. However, the two PDX shown are both lacking important control groups. For PC_105 anti-LIF treatment is lacking, while for PDX_187 Gem and Gem+aLIF is missing.
- Fig. 4c: Does subcellular mean nuclear expression? Presumably t-YAP?
- Fig 4f: Ab mono control is missing
- Fig. S4b: this is unfortunately the only figure where the authors provide IHC data, which are not informative at all. In my view, this is a bit disappointing and clearly limits the otherwise exciting findings as it is very difficult to get a biological understanding for LIF expression and dependent pathways such as YAP from both the tumor and stromal compartment. To extend the importance of this potentially highly relevant signaling axis with regard to compartment-specific (tumor vs. stroma) signaling cues and also for its value as biomarker (as discussed by the authors (pg. 8), the authors should perform IHC or multiplex IF for LIF and YAP in their syngenic mouse models and the PDX and also in a large series of human PDAC (e.g. a TMA) to analyze the frequency, tumor/stroma expression pattern and correlation of the LIF-YAP axis. This analysis should be presented in the main figures.

Reviewer #3 (Remarks to the Author):

The authors have made a significant effort to address the concerns of all reviewers and as a consequence the manuscript has improved. One major flaw still remains, that being the lack of insight as to how KRAS regulates LIF. Despite this, the manuscript is suitable for publication in Nature Communications. However, there are a few other minor issues that should be addressed, as outlined below.

1. In Fig 1a shKRAS #60 and shKRAS #62 are used in 3 of the 6 cell lines. Only shKRAS #62 is used in the remaining cell lines. Both hairpins should be used in all cell lines.
2. There is inadequate statistical analysis performed in Fig S1a.
3. Panc1.0 pLKO.1 cells should be included in this experiment. As it currently stands, the experiment is meaningless as the reader is not provided information as to basal levels of p-STAT3 or the effect of LIF supplementation on p-STAT3 in Panc1.0 cells. Even more problematic is the fact that LIF does not stimulate STAT3 phosphorylation in the context of shKRAS #62. This experiment should be carried out in some of the other cell lines utilised in Fig 1a.
4. The statement in lines 75-76 ("Adding LIF in the culture media...") is inaccurate. LIF did not rescue growth in 2D in the context of GSK1120212 treatment, and LIF did not appreciably affect growth in 3D in the context of either GSK1120212 or PD0325901.
5. In Fig 2a the authors employ shKRAS #59, which is not used in Fig 1a. For the purpose of consistency, shKRAS #60 and shKRAS #62 should be employed in Fig 2a or shKRAS #59 should be included in Fig 1a. Also, the extent of KRAS knockdown is not validated for shKRAS #59
6. The statement in lines 82-84 ("Increased expression of the stem cell markers...") is not true. IL6 did stimulate an increase in CD44 expression in 2 of the 3 cell lines and an increase in abcb1 in 1 of the 3 cell lines.
7. The authors still do not adequately address why LIF antibody is effective as a single agent in the xenograft model (Fig 3c) but not the syngeneic model (Fig 3d).
8. To complement the luciferase assays performed in Fig 4a the authors should examine changes in

expression of bona fide YAP target genes (e.g. CTGF, ANKRD1, etc.).

9. In Figure 4g, why do the authors switch between YAP siRNA and shRNA in the different cell lines? More than one shRNA should be employed or rescue experiments should be performed.

Reviewer #1 (Remarks to the Author):

The authors have significantly improved their manuscript in the revised version. Important changes include the addition of more cell lines, two PDX models for the in vivo experiments. and additional knockdown experiments interfering with YAP/TAZ TEAD that suggest a role for this pathway in LIF-mediated signaling, at least in a sphere forming assay. A number of additional improvements have been made to the manuscript to address the reviewers' comments. This includes a much better description of the experimental procedures, improved clarity of the figures and additional minor changes. In general, this is an interesting manuscript in an important field and the experiments appear to support the conclusions. I would recommend publication of the manuscript once the remaining points are addressed. The main point I still consider insufficiently addressed is the description of the statistical parameters used. This omission makes interpretation of some of the experiments difficult or impossible.

Statistical parameters:

Despite having ticked all 'confirmed' boxes on the Reporting Summary form, the authors insufficiently not at all or describe a number of important statistical parameters important for interpretation of the results. Maybe the authors could enlist the help of someone experienced in biostatistics and re-evaluate all figures that include (or should include) statistical testing?

Examples are:

A) The authors should state whether the 'N' indicated refers to biological or technical replicates in cases where this is not clear from the description.

B) The tests used should be indicated and whether they are one- or two-sided.

C) Still, the error bars are not clearly defined.

D) A full description of the statistics including central tendency (e.g. means) or other basic estimates (e.g. regression coefficient) AND variation (e.g. standard deviation) or associated estimates of uncertainty (e.g. confidence intervals) is missing.
and/or

E) For null hypothesis testing, the test statistic (e.g. F, t, r) with confidence intervals, effect sizes, degrees of freedom and P value noted

Response: The Online method and figure legends have been modified to clarify the statistical analysis.

Reviewer #2 (Remarks to the Author):

This manuscript, which has been previously reviewed for [redacted] has been significantly improved with more in vivo models and genetic YAP inactivation.

In principal I think that this work is suitable for acceptance, however, some issues remain.

- Fig. 1d+e: the legends got mixed up, please revise. The “Oncomine analysis” remains ill-described and should include more details to follow it.

Response: The legends of Fig 1 d-f have been corrected. More information about “Oncomine analysis” has been added in the legend of Fig 1d.

- Fig. 3g: the authors explicitly state (pg. 6, line 126) that anti-LIF was tested in PDXs. However, the two PDX shown are both lacking important control groups. For PC_105 anti-LIF treatment is lacking, while for PDX_187 Gem and Gem+aLIF is missing.

Response: The treatments of anti-LIF antibody alone, gemcitabine, or gemcitabine with or without anti-LIF antibody were carried out in PDX_PC187. The tumor growth has been demonstrated in updated Fig 3g. The basic expression level of LIF has been examined by using western blot analysis and has been demonstrated in the Supplementary Figure 3 b. The content and figure legends have been modified accordingly.

- Fig. 4c: Does subcellular mean nuclear expression? Presumably t-YAP?

Response: Yes. The nuclei were first isolated, and then the expression level of total YAP at protein was evaluated by using western blotting analysis. The figure legend and online method have been modified accordingly.

- Fig 4f: Ab mono control is missing

Response: Due to the low tumor proliferation rate of PDX_PC105, we did not have enough mice to include the group treated with anti-LIF antibody alone. To address the issue, we have stained the tumors harvested from syngeneic mouse models (Fig 3 d & e) and performed the histochemical staining for phospho-YAP at Ser127 (updated Supplementary Fig 4 e right panel). When compared to the tumor treated with gemcitabine alone, the tumors with anti-LIF antibody and/or gemcitabine showed elevated expression of P-YAP (S127). The content and figure legend have been updated accordingly.

- Fig. S4b: this is unfortunately the only figure where the authors provide IHC data, which are not informative at all. In my view, this is a bit disappointing and clearly limits the otherwise exciting findings as it is very difficult to get a biological understanding for LIF expression and dependent pathways such as YAP from both the tumor and stromal compartment.

Response: We understand and appreciate the concern. To address this issue, we have stained two panels of human pancreas tissue arrays (Biomax PA484 and PAN241) with LIF or P-YAP (S127). The images were taken, and the intensity of was quantified by the KEYENCE BZ-X800 microscope and its associated software. The new results have been added in Fig. 4h and Supplementary Fig. 4f. The content and figure legends have been updated accordingly.

Reviewer #3 (Remarks to the Author):

The authors have made a significant effort to address the concerns of all reviewers and as a consequence the manuscript has improved. One major flaw still remains, that being the lack of insight as to how KRAS regulates LIF. Despite this, the manuscript is suitable for publication in Nature Communications. However, there are a few other minor issues that should be addressed, as outlined below.

1. In Fig 1a shKRAS #60 and shKRAS #62 are used in 3 of the 6 cell lines. Only shKRAS #62 is used in the remaining cell lines. Both hairpins should be used in all cell lines.

Response: For the consistency, we understand the concern risen by the reviewer. However, most of pancreatic cancer cell lines are KRAS-dependent. Upon the stable expression of shRNA against KRAS, some of the cells show arrested proliferation and are unable to be expanded/passaged. That is the reason that one of the hairpin is lacked in some of the cell lines shown in Fig.1 a. To address this concern, we used different siRNAs to knock down KRAS in the following cell lines and confirmed that knocking down KRAS decreased their LIF expression.

2. There is inadequate statistical analysis performed in Fig S1a.

Response: More information has been added in Fig S1a and the related legend.

3. Panc1.0 pLKO.1 cells should be included in this experiment. As it currently stands, the experiment is meaningless as the reader is not provided information as to basal levels of p-STAT3 or the effect of LIF supplementation on p-STAT3 in Panc1.0 cells. Even more problematic is the fact that LIF does not stimulate STAT3 phosphorylation in the context of shKRAS #62. This experiment should be carried out in some of the other cell lines utilised in Fig 1a.

Response: Panc1.0 pLKO.1 and HcG25 cell lines have been included. Supplementary Fig 1 b along with content and figure legend have been modified accordingly.

4. The statement in lines 75-76 (“Adding LIF in the culture media...”) is inaccurate. LIF did not

rescue growth in 2D in the context of GSK1120212 treatment, and LIF did not appreciably affect growth in 3D in the context of either GSK1120212 or PD0325901.

Response: Thank you for the comment, and the content has been modified.

5. In Fig 2a the authors employ shKRAS #59, which is not used in Fig 1a. For the purpose of consistency, shKRAS #60 and shKRAS #62 should be employed in Fig 2a or shKRAS #59 should be included in Fig 1a. Also, the extent of KRAS knockdown is not validated for shKRAS #59

Response: For the consistency, the knockdown efficiency of shKRAS#59 in Panc1.0 cell line has been validated. Fig 1a has been modified accordingly.

6. The statement in lines 82-84 (“Increased expression of the stem cell markers...”) is not true. IL6 did stimulate an increase in CD44 expression in 2 of the 3 cell lines and an increase in abcb1 in 1 of the 3 cell lines.

Response: The content has been modified accordingly.

7. The authors still do not adequately address why LIF antibody is effective as a single agent in the xenograft model (Fig 3c) but not the syngeneic model (Fig 3d).

Reponses: Thank you for the comment. The purposes of the experiments shown in Fig 3c and 3d are different. As shown in Fig 3c, pre-treatment of LIF antibody can prevent the xenograft tumor initiation. In syngeneic mouse model, the treatments including anti-LIF antibody alone have been conducted after the detectable tumors formed. To eliminate the potential inferences of LIF secreted from the immune cells of the hosts in syngeneic model, we also conducted the similar experiments in PDX models (Fig 3f & g), suggesting that LIF antibody along with gemcitabine significantly repressed tumor growth when compared to gemcitabine or antibody alone.

8. To complement the luciferase assays performed in Fig 4a the authors should examine changes in expression of bona fide YAP target genes (e.g. CTGF, ANKRD1, etc.).

Response: qPCR to detect expressions of CTGF and ANKRD at mRNA has been performed in the cells treated with IL6 or LIF at 100 ng/ML for 72 hours. The result has been added as Supplementary Fig. 4b. The content and figure legend have been modified accordingly.

9. In Figure 4g, why do the authors switch between YAP siRNA and shRNA in the different cell lines? More than one shRNA should be employed or rescue experiments should be performed.

Response: For the consistency, we employed two different siRNAs to knock down *YAP1* in pancreatic cancer cell lines. The new results have been added as Fig. 4g. The content and figure legend have been updated accordingly.

We hope that we have addressed all concerns and that the paper is now acceptable for publication.
We look forward to hearing back from you.

REVIEWERS' COMMENTS:

Reviewer #1 (Remarks to the Author):

The authors have addressed all remaining points and I recommend publication.

Reviewer #2 (Remarks to the Author):

The authors have addressed my issues and the manuscript has been significantly improved.

Minor issues to be clarified:

- Fig. 4h right: I do not understand this quantification figure at all?
- Suppl. Fig. 4e right: the staining in this low magnification looks very unspecific. The authors should add an insert with higher magnification in order to acknowledge the staining pattern and signal localization.
- Suppl. Fig. 4f: the shown "lesion" looks a bit like an pancreatic islet but I suppose this is not the case.

Reviewer #3 (Remarks to the Author):

The authors have adequately addressed the concerns raised by the reviewers.

Please see point-by-point responses as following:

Reviewer #1 (Remarks to the Author):

The authors have addressed all remaining points and I recommend publication.

Reviewer #2 (Remarks to the Author):

The authors have addressed my issues and the manuscript has been significantly improved.

Minor issues to be clarified:

- Fig. 4h right: I do not understand this quantification figure at all?

Response: X axis represents IHC scores of LIF staining (ranging from 0 to 3+), and Y axis represents the scores of phospho-YAP (S127) IHC staining (from 0 to 3+). Each dot represents each sample on the human pancreatic tissue arrays, including normal tissues and cancer samples. The staining of LIF and p-YAP (S127) shows negative correlation (LIF high and p-YAP low on the same samples, vice versa) The correlation curve shown here includes 73 samples. Some of them show the same scores of LIF and p-YAP staining, and their represented dots are overlapping. The R score and two tailed P value was calculated by using GraphPad Prism 7.

- Suppl. Fig. 4e right: the staining in this low magnification looks very unspecific. The authors should add an insert with higher magnification in order to acknowledge the staining pattern and signal localization.

Response: We have enlarged the images in higher magnification and added one with positive staining in high magnification. The revised figure is shown as Fig 4e (right panel).

- Suppl. Fig. 4f: the shown "lesion" looks a bit like an pancreatic islet but I suppose this is not the case.

Response: The stage of pancreatic cancer tissue is scored and given by BioMax. We do understand the lack of detailed pathological information can be misleading. Therefore, we have removed this specific area of images.

Reviewer #3 (Remarks to the Author):

The authors have adequately addressed the concerns raised by the reviewers.